

# Technical Note: A simple theoretical model framework to describe plant stomatal "sluggishness" in response to elevated ozone concentrations

Chris Huntingford[1], Rebecca J. Oliver[1], Lina M. Mercado[2,1], and Stephen Sitch[2]

[1]Centre for Ecology and Hydrology, Benson Lane, Wallingford, Oxfordshire, OX10 8BB, U.K.
[2]College of Life and Environmental Sciences, University of Exeter, Amory Building, Rennes Drive, Exeter, EX4 4RJ, U.K.

*Correspondence to:* Chris Huntingford (chg@ceh.ac.uk)

**Abstract.** Elevated levels of tropospheric Ozone $[O_3]$ causes damage to terrestrial vegetation, affecting leaf stomatal functioning and reducing photosynthesis. Climatic impacts under future raised atmospheric Greenhouse Gas (GHG) concentrations will also impact on the Net Primary Productivity (NPP) of vegetation, which might for instance alter viability of some crops. To-

5 gether, ozone damage and climate change may adjust the current ability of terrestrial vegetation to offset a significant fraction of carbon dioxide ($CO_2$) emissions. Climate impacts on the land surface are well studied, but arguably large-scale modelling of raised surface level $[O_3]$ effects is less advanced. To date most models representing ozone damage use either $[O_3]$ concentration or, more recently, flux-uptake related reduction of stomatal opening, estimating suppressed land-atmosphere water and $CO_2$ fluxes. However there is evidence that for some species, $[O_3]$ damage can also cause an inertial "sluggishness" of stomatal

response to changing surface meteorological conditions. In some circumstances e.g. droughts, this loss of stomata control can cause them to be more open than without ozone interference. The extent of this effect may be dependent on magnitude and cumulated time of exposure to raised $[O_3]$, suggesting experiments to analyze this require operation over long timescales such as full growing seasons. To both aid model development and provide empiricists with a system on to which measurements can be mapped, we present a parameter-sparse framework specifically designed to capture sluggishness. This contains a single

time-delay parameter $\tau_{O_3}$, characterising the timescale for stomata to catch up with the level of opening they would have without damage. The larger the value of this parameter, the more sluggish the modelled stomatal response. Through variation of $\tau_{O_3}$, we find it is possible to have qualitatively similar responses to factorial experiments with and without raised $[O_3]$, when comparing to measurement timeseries presented in the literature. This low-parameter approach lends itself to the inclusion of ozone-induced inertial effects being incorporated in the terrestrial vegetation component of Earth System Models (ESMs).

## 1 Introduction

Anthropogenic emissions from industrial processes, transport and biomass burning are increasing background levels of surface ozone $[O_3]$ (mol mol$^{-1}$) (Vingarzan, 2004). There is much evidence this adjusts the stomatal opening of terrestrial vegetation, and so influencing land-atmosphere exchanges of water and carbon both globally and locally (Ainsworth et al., 2012; Wittig



et al., 2007, 2009; Mills et al., 2016). This may reduce the ability of vegetation to photosynthesize, which at the global scale is a concern as it may lower the current fraction of $CO_2$ emissions the land draws down (Felzer et al., 2005; Sitch et al., 2007; Lombardozzi et al., 2015). At more local-to-regional scales, ozone-induced damage could affect crop yields and potentially food security (Ainsworth et al., 2012; Anav et al., 2011; Avnery et al., 2011; Tai et al., 2014).

Increasingly though, for some plant species the situation is discovered to be more complex. A growing number of species are found to show increased stomatal opening and/or delayed stomatal opening, termed stomatal sluggishness, is caused by raised concentrations of ozone (Mills et al., 2016). Under stressed conditions, such as drought, the mechanism has been linked to ozone interfering with the hormonal signalling pathway abscisic acid (ABA) (Wilkinson and Davies, 2009, 2010; Mills et al., 2009). ABA is used by plants to communicate to stomata the need to reduce opening in the presence of growing abiotic stress

conditions. Specifically, elevated ozone stimulates ethylene production which prevents ABA from otherwise closing stomata (Wilkinson and Davies, 2009, 2010). Loss of stomatal control is observed in response to a range of environmental factors, including drought (Wilkinson and Davies, 2009, 2010; Mills et al., 2009; Hayes et al., 2012; Wagg et al., 2013), high light (Paoletti and Grulke, 2010; Hoshika et al., 2012b; Wagg et al., 2013) and high vapour pressure deficit (Grulke et al., 2007). The ABA signalling pathway mediates stomatal responses to many of these stress factors, as has been found particularly in drought

conditions. It is therefore likely to play a role in controlling stomatal responses to ozone under fluctuating environmental conditions.

    Ozone-induced sluggishness can have the opposite effect to that generally associated with $O_3$ damage. In some circumstances stomata are more open than without $O_3$ influence. Ozone-induced sluggish behaviour that delays stomatal closure means affected plants create a positive feedback whereby they receive a higher $O_3$ flux with greater $O_3$ damage resulting. Impacted

plants could also lose more water, and if this occurs during drought episodes for example, this may exacerbate soil moisture deficits, in turn affecting NPP. Hence there are implications for water use, crop yields and food security (Sun et al., 2012; Tai et al., 2014; Van Dingenen et al., 2009). It is noted that for many regions (e.g. Europe), high tropospheric $O_3$ events often correspond to atmospheric blocking anticyclones, themselves frequently linked to drought events.

    At the regional scale, McLaughlin et al. (2007a, b) and Sun et al. (2012) provide field evidence of increased transpiration and

25 reduced streamflow in forests. This is attributed to a sluggish stomatal response to ambient levels of $O_3$. This could increase the frequency and severity of droughts, suppress forest productivity and add to any direct $O_3$ inhibition of photosynthetic capacity. However, in contrast, Hoshika et al. (2012a) found that despite sluggish stomatal control in trees exposed to $O_3$, whole tree water use reduced due to lower gas exchange and premature shedding of injured leaves. The literature suggests that sluggish stomata response to $O_3$ is not ubiquitous (Mills et al., 2016; Wittig et al., 2007); which species respond this way and under what

conditions requires understanding. For species affected, significant impacts on watershed hydrology and carbon sequestration are possible.

    Most large-scale terrestrial models represent raised tropospheric ozone concentrations as detrimental to photosynthesis, inducing extra stomatal closure (Wittig et al., 2007). For instance, the JULES (Joint UK Land Environment Simulator) model uses a flux-gradient approach to describe simulated plant $O_3$ damage (Sitch et al., 2007; Clark et al., 2011). The model is

35 parameterised to reduce photosynthesis in response to accumulated $O_3$ uptake, and because in JULES this processes is coupled



to stomatal conductance, that also decreases. This has similarities to how ozone damage representation has been introduced by Franz et al. (2017) to the OCN land model (Zaehle and Friend, 2010). Lombardozzi et al. (2012), for the CLM (Community Land Model), decouple photosynthesis and stomatal conductance so that raised surface $O_3$ levels reduce carbon assimilation disproportionately more than transpiration. A first attempt to numerically emulate the sluggish feature of higher stomatal

opening is by Hoshika et al. (2015). They modulate the multi-layer atmosphere-soil-vegetation (SOLVEG) terrestrial model so the minimum stomatal opening in the Ball-Woodrow-Berry model, $g_{\min}$ (m s$^{-1}$), increases for higher cumulative $O_3$ exposure. This potentially raises transpiration losses.

Geographically-extensive projections of ozone impacts on the land surface response need understanding within the context of other large-scale changes affecting terrestrial ecosystems. These include the direct physiological effect of raised $CO_2$ through

fossil fuel burning, the impact of climate change due to raised $CO_2$ and other GHGs, and aerosols adjusting the composition of downward shortwave radiation (Huntingford et al., 2011). Even if an emissions trajectory is followed that achieves global warming stabilised at 2°C above pre-industrial levels, general near-surface warming over land will be higher (Huntingford and Mercado, 2016). Therefore even moderate levels of global warming could have strong influences on terrestrial vegetation, and in this situation any additional ozone-induced changes need to be described. Earth System Models (ESMs) are the main

tools to describe the effect on climate of raised atmospheric GHGs, and interactions and feedbacks on global biogeochemical cycles. Such models contain a land surface component, e.g. the JULES model (Clark et al., 2011) within the HadGEM2-ES ESM (Jones et al., 2011). HadGEM2-ES ESM carries ozone as an atmospheric tracer, to which JULES responds (Sitch et al., 2007). ESMs contribute to global model databases, and most recently the fifth phase of the Coupled Model Intercomparison Project, CMIP5 (Taylor et al., 2012), which inform the United Nations Intergovernmental Panel on Climate Change reports e.g.

IPCC (2013). If a substantial fraction of vegetation responses to elevated tropospheric ozone contain stomata sluggishness, this requires implementation in large-scale terrestrial vegetation models and ESMs to assess global implications. Any influence on terrestrial carbon stores is important for attribution and understanding of recent trends in the land carbon sink e.g. (Le Quere et al., 2018).

Opportunities exist to incorporate inertia within mechanistic equations. Direct ozone interactions with abscisic acid may be

modelled, if a suggestion is fulfilled that the ABA hormone be included in large-scale land models (Huntingford et al., 2015). However to proceed before then, a more empirically-based description is required. By definition, stomatal sluggishness implies a timescale exists, describing the delay behind a state without ozone damage. We call this timescale $\tau_{O_3}$ (s).

## 2    Sluggishness parameter $\tau_{O_3}$ and modelled stomatal opening

Proposed is a simple and minimal mathematical description of sluggishness. We first set the time-evolving leaf-level stomatal

opening that would occur without ozone damage as $g_l(t)$ (m s$^{-1}$). This is assumed to respond to the standard drivers of temperature $T$ (K), light level i.e. photosynthetic active radiation $I_P$ (W m$^{-2}$), vapour pressure deficit VPD (kPa) and soil moisture status $\theta$ (kg water (kg soil)$^{-1}$). A second variable is defined as the stomatal opening with additional ozone-induced sluggishness and named $g_{l,\mathrm{slug}}(t)$ (m s$^{-1}$). Sluggishness is characterised by a single new parameter $\tau_{O_3}$ (s), representing



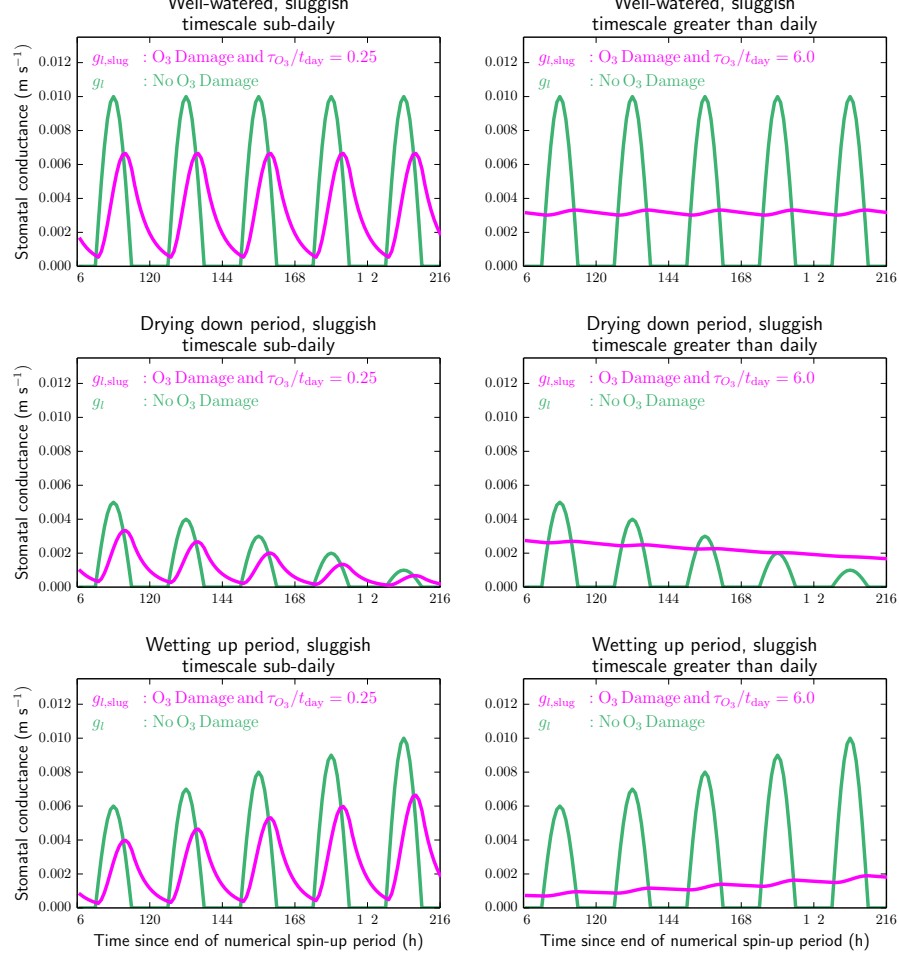

**Figure 1.** The effect of low (left-hand panels) and high (right-hand panels) levels of stomatal sluggishness. Calculations for stomatal conductance correspond to well-watered conditions (top row), entering a period of drought (middle row) and recovery from drought (bottom row). Simulations are for five 24-hour periods. Green curves are stomatal conductance without ozone effects, and magenta curves are with sluggishness. Appendix A details the modelling framework and driving conditions leading to these curves

the timescale of how long ozone-damaged stomata take to "catch up" with the level of opening without $O_3$ influence i.e. $g_l = g_l(T, I_P, \mathrm{VPD}, \theta)$. This leads to the ordinary differential equation in time $t$ (s) of:

$$\frac{\mathrm{d}g_{l,\mathrm{slug}}}{\mathrm{d}t} = -\frac{g_{l,\mathrm{slug}} - g_l}{\tau_{O_3}}. \tag{1}$$

For this technical note, two illustrative sets of solutions to Eq. (1) are considered. Setting $t_{\mathrm{day}} = 86400$ (s) as the number of
5  seconds in a day, first sluggishness effects for a timescale less than one day, of $\tau_{O_3}/t_{\mathrm{day}} = 0.25$ are modelled. Then a second set, corresponding to a more sluggish timescale that is significantly greater than one day are considered, with $\tau_{O_3}/t_{\mathrm{day}} = 6$.



These are shown, respectively, as the magenta curves in the left-hand and right-hand columns of Fig. 1. The green curves are with no $O_3$ sluggish damage, showing identical curves for $g_l$ between the two diagram columns. The background "sluggish-free" curves for $g_l$ are described in the Appendix, and they broadly correspond to three cases (for each of the two columns) as daily variability for: (i) well-watered vegetation, (ii) a period of increasing drought conditions and (iii) recovery from drought.
These correspond to the top, middle and bottom rows respectively of Fig. 1.

The simulations are summarised as follows. In the well-watered case (top row), for $\tau_{O_3}/t_{\mathrm{day}} = 0.25$ there remains a sizeable diurnal cycle in the ozone-damaged stomatal conductance $g_{l,\mathrm{slug}}$. For $\tau_{O_3}/t_{\mathrm{day}} = 6$, almost all within-day variation is lost and stomata remain open throughout the nighttime periods. For drying conditions (middle panels), again for the smaller $\tau_{O_3}$ case, there remains subdiurnal variability, and the downward trend is similar between damaged and undamaged stomata. However,
for larger $\tau_{O_3}$, the solution to Eq. (1) is such that the larger inertia makes stomata eventually more open than at any point during the diurnal cycle of those that are undamaged. This scenario is starting to receive particular interest, with emerging evidence that ozone damage can under some circumstances cause excessive opening of stomata. In the bottom row, the lower $\tau_{O_3}$ example (left) shows again delays at subdiurnal timescale, but the damaged stomata retain capability to open more as conditions become more favourable. For the higher $\tau_{O_3}$ case, there is only minimal ability to keep up with increases in opening
by the modelled undamaged stomata.

## 3   Discussion

There is evidence in the literature that some features of Fig. 1 can be seen in measurements. Top left panel (well-watered, low $\tau_{O_3}$) has strong similarities to Fig. 2 of Hoshika et al. (2012b). In that study, a type of beech tree is subjected to control experiments, with ambient and enhanced ozone concentrations. Similar curves are seen for beans, in Fig. 1 of Paoletti and
Grulke (2010). Both studies are for relatively short exposure times of the vegetation subjected to higher $O_3$ levels. Data-based analysis of grassland species by Hayes et al. (2012) assesses the effects of different ozone concentration exposures, for both well-watered and drought conditions. In that paper, their Fig. 4c shows that after 19 weeks and for the highest ozone exposure levels, the stomata are roughly as open in the drought conditions as for the well-watered circumstances. This near-complete loss of ability to respond to emerging drought conditions has similarities to middle row, right-hand panel of our Fig. 1.
Our mathematical framework of Eq. (1) and solution for two representative $\tau_{O_3}$ values, raises a set of conjectures, issues and questions about the implications of stomatal sluggishness. This can aid developing future measurement campaigns of ozone effects on stomatal conductance, to test the validity of Eq. (1) and then its parameterisation if verified as an appropriate model.

For sluggishness at sub-daily periods $\tau_{O_3}/t_{\mathrm{day}} < 1$, stomatal conductance $g_{l,\mathrm{slug}}$ has some symmetry, with periods of both larger and smaller opening, when compared to $g_l$. However, this may cause an asymmetry for photosynthetic activity, as
there are periods at night when sluggish stomata are open (left columns of Fig. 1) and when photosynthesis would not occur. Conversely daytime opening is suppressed in $g_{l,\mathrm{slug}}$, and so overall sluggish stomatal response will lower terrestrial carbon uptake. This is seen in Fig. 2b of Hoshika et al. (2012b). Hence when averaged over periods longer than one day, sluggishness will dampen overall draw-down of atmospheric $CO_2$. This could cause a mis-attribution of effect, if measurements are made





during daytime light periods only and with $\tau_{O_3}/t_{\text{day}} < 1$. This is because in the presence of stomatal sluggishness, and with measurements made only when stomata are less open than without $O_3$ damage (i.e. no night measurements), it could be inferred that the more conventional non-sluggish descriptions of damage are valid. An open research question is whether stomata could have both responses. That is the commonly modelled ozone flux-based (or concentration-based) description that always reduces

stomatal opening, as well as an additional inertial contribution.

With evidence that ozone damage can cause raised stomatal opening, in some circumstances and for some species, this is of concern during periods of approaching drought, high temperatures or both. Under severe ozone damage with $\tau_{O_3}/t_{\text{day}} \gg 1$ and during "drying down" periods, raised evapotranspiration through larger stomatal opening could trigger severe water stress. This may deplete soil moisture to levels that would not otherwise have been attained. This could cause wilting, or initiate plant

hydraulic failure through embolism or cavitation, with clear implications for crop viability and food security in regions that experience seasonal drought.

Observational evidence of different levels of sluggishness suggests these are a function of accumulated exposure e.g. Hoshika et al. (2015). For existing models of $O_3$ damage to stomata, a level exists and only above which damage occurs, to account for the ability of vegetation to detoxify low levels of ozone. In Sitch et al. (2007) for instance, that threshold is a level of

ozone flux in to vegetation. Together this implies that the evolution of $\tau_{O_3}$, possibly dependent on time since the start of the growth season, $t_{g,\text{start}}$ (s), can be described by two parameters. The first is a critical threshold above which damage occurs, as flux $F_{O_{3,\text{crit}}}$ (nmol m$^{-2}$ s$^{-1}$) (or concentration [$O_{3,\text{crit}}$] (mol mol$^{-1}$)). The second linearly relates time spent over the threshold to the amount of sluggishness, expressed by changes to $\tau_{O_3}$. Hence $\tau_{O_3}(t) = b\int_{t_{g,\text{start}}}^{t} \max[F_{O_3} - F_{O_{3,\text{crit}}}, 0]\mathrm{d}t$ or $\tau_{O_3}(t) = b\int_{t_{g,\text{start}}}^{t} \max[[O_3] - [O_{3,\text{crit}}], 0]\mathrm{d}t$. This second parameter $b$ has units of either s[nmol m$^{-2}$]$^{-1}$ or [mol mol$^{-1}$]$^{-1}$

respectively.

If the ABA signalling process plays a key role in linking tropospheric ozone levels to stomata sluggish effects, then needed is careful analyses of data from experimental examples of well-watered vegetation, and with high $O_3$ levels. ABA concentrations increase during periods of soil moisture stress, to which stomata respond by lowering their opening. Any presence of sluggishness during well-watered periods, and thereby for low ABA concentrations, suggests that additional mechanisms operate

beyond this hormone in linking ozone concentrations to inertial effects of stomata.

## 4  Conclusions

We present a simple first-order differential equation to characterise the observed "sluggish" response of modelled stomata to elevated levels of tropospheric ozone. The formulation is deliberately parameter-sparse, with a single parameter $\tau_{O_3}$. This parameter represents a delay, characterising the timescale required for ozone-damaged stomata to "catch up" with the value it

would have without ozone-induced damage.

Through simple numerical examples we illustrate how, depending on circumstances, this equation can project stomata to be both more closed than they would otherwise be, and critically the opposite whereby sluggishness can provide a mechanism for additional opening. Stomata that are more open through ozone damage has been reported from observations, yet is not





currently routinely included in land surface response models. This is because most existing modelling schemes can only lower stomatal opening for raised $O_3$ levels.

Targeted measurement campaigns may provide more detailed information on the appropriateness of our $\tau_{O_3}$ formulation. This includes (a) whether this is a generic form for describing tropospheric ozone damage to vegetation, (b) its value and

5 variations due to accumulated ozone exposure, and (c) if there is potential to map on to broad plant functional types. If the latter is possible, then "sluggish" effects can be implemented within large-scale land surface models such as JULES (Clark et al., 2011). To numerically solve Eq. (1), an additional variable of stomatal conductance $g_{l,\text{slug}}$ at the previous timestep must be carried for each modelled gridbox.

Any additional stomata opening through ozone damage is of concern for regions experiencing drought. Further ozone-

10 induced transpiration could dry soils sufficiently that plant wilting is either reached earlier, or would otherwise have been avoided without $O_3$ damage. This has food security implications if affecting crops. Furthermore, long-term (i.e. chronic) ozone influence on photosynthetic capability may alter terrestrial carbon stores. Uncertainty in the modelled global carbon cycle is almost as large as that of the physical climate in terms of predicting expected future warming levels (Huntingford et al., 2009). If calibrated and implemented in large-scale terrestrial models and then ESMs, our formulation offers hope that the implications

of sluggish stomata can be understood in the context of simultaneous changing climatic conditions, the global carbon cycle and varying tropospheric ozone levels.

## 5    Code availability

Python code leading to Fig. 1 is available on request from C.H. (chg@ceh.ac.uk)

## Appendix A:  Parameters leading to illustrative Fig. 1

The driving conditions leading to the illustrative simulations of Fig. 1 are as follows. In well-watered conditions, and without ozone damage influence, a daily maximum stomatal opening $g_{l,\text{max}}$ is assumed invariant, at 0.01 m s$^{-1}$. This is representative of midday values, under high sunlight levels and with well-watered conditions. This corresponds to the top row of Fig. 1. Sub-daily variability is then described as the part of a sinusoidal function when positive, as:

$$g_l = g_{l,\text{max}}(t) \times \max\left\{-\cos\left(\frac{2\pi t}{t_{\text{day}}}\right), 0\right\}. \tag{A1}$$

"Drying down" is represented by changing $g_{l,\text{max}}$ on a daily basis, following a period of being well-watered at 0.01 m s$^{-1}$. This occurs over nine days, down to a minimum stomatal opening of 0.001 m s$^{-1}$, falling by 0.001 m s$^{-1}$ each day. This is the middle row of Fig. 1. "Wetting up" is described as following a period with low conductance of 0.001 m s$^{-1}$, rising to 0.01 m s$^{-1}$ over nine days, and so the bottom row of Fig. 1. These calculations of $g_l$ are the green curves throughout the diagram.

Equation (1) is then solved to calculate $g_{l,\text{slug}}(t)$, for the corresponding values in each diagram panel of $g_l$. This is with left

panels of $\tau_{O_3}/t_{\text{day}} = 0.25$ and right panels of $\tau_{O_3}/t_{\text{day}} = 6.0$. As Eq. (1) is a non-equilibrium solution, then initial conditions





are required. We do this numerically, by "spinning up" over 100 repeated initial days, which in the top and middle panels are well-watered with $g_{l,\max} = 0.01$ m s$^{-1}$ and bottom panels is drought conditions with $g_{l,\max} = 0.001$ m s$^{-1}$.

*Author contributions.* C.H. created the theoretical model, operated the numerical experiments and designed the paper. R.O., L.M.M. and S.S. performed the literature review and placed the analysis in the context of existing research. All authors wrote the paper.

5 *Competing interests.* The authors confirm they have no competing interests.

*Acknowledgements.* C.H. R.J.O. and L.M.M. acknowledge support from the NERC-CEH National Capability Fund. R.J.O. and L.M.M. acknowledge support from U.K. Natural Environment Research Council grant NE/N017951/1. L.M.M. and S.S. acknowledge support from U.K. Natural Environment Research Council grant NE/R001812/1.



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
