# Peer review of "Technical Note: A simple theoretical model framework to describe plant stomatal "sluggishness" in response to elevated ozone concentrations"

_Biogeosciences, 2018_

## Referee Comment (RC1) · Anonymous Referee #1 · 13 Jun 2018

The authors present a convincing and thoroughly justified and explained simple theoretical approach for the calculation of stomatal sluggishness following plants' exposure to ozone for consideration in ESMs. It remains to be seen whether this phenomenon is of significance for the modelling of the global carbon cycle (the impact of ozone on vegetation is unquestionable, but much more experimental evidence is needed to demonstrate that ozone-induced sluggishness occurs across various vegetation types, species and indeed cultivars under various climatic conditions) and the suggested approach might in the long run be considered too simplistic if a more sophisticated biochemical approach for the effect of ozone on abscisic acid and the inclusion of ABA in ESMs will become available, but for the time being - i.e. as an interim solution -

the presented model framework is a thought-provoking suggestion that will further the respective scientific debate and enable modellers to potentially add more precision to the calculation of ozone effects on the terrestrial carbon balance. Hence, I strongly support the publication of this technical note.

Three minor comments (of formal nature):

Page 2, line 6: delete "is" Page 3, line 22: e.g. should be within the following brackets Page 4, equation 1: describe what "d" in dgl and dt stands for

―――――――――――――――――――

---

## Referee Comment (RC2) · Anonymous Referee #2 · 18 Jun 2018

General comments

The Authors present a well-defined theoretical framework for inclusion of the effect of ozone-induced stomatal sluggishness in modelling systems. The concept is based on the results of empirical studies showing that the stomatal response to environmental conditions such as drought is sometimes delayed when plants are or have been exposed to ozone. The authors clearly explain and present references showing the importance of implementing this effect in modelling systems that aim to estimate the integrated effects of climate and tropospheric ozone on e.g. terrestrial carbon and water cycles. The framework is presented with the aim of aiding model development and

to provide empirical studies with relevant parameters for mapping their results. For this purpose the presented framework is interesting and useful. For the modelling community, the authors present a timely initiative for further development. However, given that the authors have not yet validated the presented the concept towards empirical data, no suitable parameter values are presented, which means that the framework in its current form serves more as a starting point for further development for the modelling community. The Authors do not suggest how the variations in sluggishness across plant species/plant functional types should be implemented in models, but do point out the need for empirical data for this. Given the limited data available for testing and validating the presented framework, the approach in its current form largely based on assumptions about relevance across plant types/ecosystems and as the Authors highlight, needs to be further developed before being implemented in models. However, as a first step towards a fully developed parameterisation the authors present an interesting approach with the potential to enhance future estimates of the effects of ozone induced damage to terrestrial vegetation.

Specific comments

P.1 lines 11-13: Consider moving this to the introduction.

P. 2, lines 22-23: Please explain the link between ozone events and blocking anticyclones, or insert reference.

P. 4, Eq. (1): The application of the sluggishness parameter is illustrated with two example values that are shown to fit with some selected results in the literature. Although interesting, the selected parameter values seem somewhat arbitrary, and are not linked to other factors presented as being relevant, such as ozone levels (as ambient concentrations or accumulated fluxes) or vegetation types. The examples would be more relevant to the community if they were put in context with corresponding conditions.

Please provide reference or other explanation for the choice of parameter values in these examples. Are they particularly relevant for certain types of vegetation? How do

these values link to ozone levels? -Have these specific values been chosen based on the empirical data sets in the literature to which they are compared in the discussion? Please provide some context and/or reference for these example values in order to make them more relevant for the readers.

P. 6, lines 16-20: The concept of linking the sluggishness to accumulated exposure to ozone has already been explained, and if necessary can easily be explained again without the use of equations in the text. Mathematical expressions and integrals are not required at this point, but could rather be introduced earlier in the paper, for example in Chapter 2.

P.7, lines 9-13: Repetition, and strictly not conclusions based on results of this article. Perhaps better placed under discussion.

Technical corrections

P. 2, line 6: remove "is"

P.3, line 18: remove "and"?

P. 6, line 17: Remove parenthesis around the concentration based critical threshold.

P. 6, line 21-22: please rephrase for clarity.

---

## Author Comment (AC2) · 12 Jul 2018

Please find below a link to our combined response to Reviewer 1 and Reviewer 2.

Please also note the supplement to this comment:
https://www.biogeosciences-discuss.net/bg-2018-206/bg-2018-206-AC2-supplement.pdf

———————————————————

---

## Author Response (AR1)

We thank BioGeoSciences for considering our short Technical Note, for sourcing two reviewers, and providing editorial advice.

Below is our responses to reviewers. The majority of this document repeats that placed on the BioGeoSciences website as Author Comment (**AC1/2**). At the end is our new and additional changes in response to the Associate Editor Report.

Please find the reviewer and editor requests in black, and with our responses below and in blue.

**Reviewer One**

The authors present a convincing and thoroughly justified and explained simple theoretical approach for the calculation of stomatal sluggishness following plants' exposure to ozone for consideration in ESMs. It remains to be seen whether this phenomenon is of significance for the modelling of the global carbon cycle (the impact of ozone on vegetation is unquestionable, but much more experimental evidence is needed to demonstrate that ozone-induced sluggishness occurs across various vegetation types, species and indeed cultivars under various climatic conditions) and the suggested approach might in the long run be considered too simplistic if a more sophisticated biochemical approach for the effect of ozone on abscisic acid and the inclusion of ABA in ESMs will become available, but for the time being - i.e. as an interim solution - the presented model framework is a thought-provoking suggestion that will further the respective scientific debate and enable modellers to potentially add more precision to the calculation of ozone effects on the terrestrial carbon balance. Hence, I strongly support the publication of this technical note.

> Thank you for these encouraging comments. We hope that our ozone-induced sluggishness "τ" description can be a useful interim description to map on to future observational campaigns that find this inertia effect.

Three minor comments (of formal nature):

Page 2, line 6: delete "is"

> Corrected

Page 3, line 22: e.g. should be within the following brackets

> Correct

Page 4, equation 1: describe what "d" in dgl and dt stands for

> Eqn 1 is an ordinary differential equation, and so the "d" follows the calculus notation. However, we now write "This leads to the ordinary differential equation, for the rate of change of gl,slug with respect to time, as:"

**Reviewer Two**

The Authors present a well-defined theoretical framework for inclusion of the effect of ozone-induced stomatal sluggishness in modelling systems. The concept is based on the results of empirical studies showing that the stomatal response to environmental conditions such as drought is sometimes delayed when plants are or have been exposed to ozone. The authors clearly explain and present references showing the importance of implementing this effect in modelling systems that

aim to estimate the integrated effects of climate and tropospheric ozone on e.g. terrestrial carbon and water cycles. The framework is presented with the aim of aiding model development and to provide empirical studies with relevant parameters for mapping their results. For this purpose the presented framework is interesting and useful. For the modelling community, the authors present a timely initiative for further development. However, given that the authors have not yet validated the presented the concept towards empirical data, no suitable parameter values are presented, which means that the framework in its current form serves more as a starting point for further development for the modelling community. The Authors do not suggest how the variations in sluggishness across plant species/plant functional types should be implemented in models, but do point out the need for empirical data for this. Given the limited data available for testing and validating the presented framework, the approach in its current form largely based on assumptions about relevance across plant types/ecosystems and as the Authors highlight, needs to be further developed before being implemented in models. However, as a first step towards a fully developed parameterisation the authors present an interesting approach with the potential to enhance future estimates of the effects of ozone induced damage to terrestrial vegetation.

We thank reviewer 2 for their very thoughtful advice. We appreciate the broadly supportive assessment, but also note some concerns. We have address these as follows:

Reviewer query: "However, given that the authors have not yet validated the presented the concept towards empirical data, no suitable parameter values are presented, which means that the framework in its current form serves more as a starting point for further development for the modelling community". This links to strongly to the comment P4 below "The application of the sluggishness parameter…..".

We try to answer this, at least in part, through much better and clearer reference to elevated experiments reported for Siebold's beech (Hoshika et al., 2012), and for grassland (Hayes et al., 2012). In the former, after approximately two months of double ambient ozone concentrations, then imposed oscillations of light levels on timescale order hours cause a slight lag in stomatal opening, compared to equivalent experiments at ambient $O_3$ levels. This is analogous to our smaller $\tau$ of sub-daily magnitude.

In the Hayes grassland experiments, analysis is made of well-watered and reduced-watered experiments, and for different $O_3$ concentration treatments. The notable feature is that for very high $O_3$ levels (order 90 nmol/mol), then beyond 9 weeks and in the drought-induced case, the stomata are almost as wide open as the well-watered case. This suggests a long-term broad inability to respond to changing conditions, and so in-keeping with our $\tau$ value of much greater than one day. Please see below as to our precise changed wording.

In response to "The Authors do not suggest how the variations in sluggishness across plant species/plant functional types should be implemented in models", we note that land surface models are evolving and starting to include more PFTs. We now write, and in part adopting reviewer wording: "The representation of general Plant Functional Types (PFTs) in land surface models is evolving, and including a larger set of them (e.g. Harper et al., 2016 changes the JULES model from five basic PFTs to nine). In the event that comprehensive measurements show variations in sluggishness between species, then this could inform future PFT definitions".

Based on this reviewer's query about PFTs, we additionally add in Discussion: "Existing PFTs in large-scale land models may have to be split to accommodate different responses. For

trees for example, birch and oak are found to have high and low sensitivity respectively in existing models of ozone-induced stomatal closure (Sitch et al., 2007). Hoshika et al. (2018)* find similarly that sluggishness effects might be stronger in white birch than deciduous oak."

\* Hoshika et al., 2018 "Ozone-induced stomatal sluggishness changes stomatal parameters of Jarvis-type model in white birch and deciduous oak", *Plant Biology*, **20**, p20-28.

Specific comments

P.1 lines 11-13: Consider moving this to the introduction.

Yes, we have done this

P. 2, lines 22-23: Please explain the link between ozone events and blocking anticyclones, or insert reference.

This is an area of active research in the UK at the moment (and undoubtedly elsewhere). However we have just checked, and as yet, the research is not yet available in peer-review papers. On this basis, we have decided to remove this sentence.

P. 4, Eq. (1): The application of the sluggishness parameter is illustrated with two example values that are shown to fit with some selected results in the literature. Although interesting, the selected parameter values seem somewhat arbitrary, and are not linked to other factors presented as being relevant, such as ozone levels (as ambient concentrations or accumulated fluxes) or vegetation types. The examples would be more relevant to the community if they were put in context with corresponding conditions.

Please provide reference or other explanation for the choice of parameter values in these examples. Are they particularly relevant for certain types of vegetation? How do these values link to ozone levels? -Have these specific values been chosen based on the empirical data sets in the literature to which they are compared in the discussion? Please provide some context and/or reference for these example values in order to make them more relevant for the readers.

Here we present the additional text now placed in the manuscript. This also gives our precise answer to the general comment by Reviewer 2 of: However, given that the authors have not yet validated.

We now write in the paper: "Our two representative values are guided by the experimental measurements presented for Siebold's beech (Hoshika et al., 2012), and for grassland (Hayes et al., 2012). In the former, after approximately two months of double ambient ozone concentrations, then imposed oscillations of light levels on timescale order hours cause a slight lag in the beech stomatal opening, compared to equivalent experiments at ambient $O_3$ levels. This is analogous to our smaller $\tau$ of sub-daily magnitude. In the grassland experiments, analysis is made of well-watered and reduced-watered experiments, and for different $O_3$ concentration treatments. The notable feature in those experiments is that for very high $O_3$ levels (order 90 nmol/mol), then beyond nine weeks and in the drought-induced case, the stomata are almost as wide open as the well-watered case. This suggests a long-term broad inability to respond to changing conditions, and so in-keeping with our high sluggish $\tau$ value of much greater than one day".

P. 6, lines 16-20: The concept of linking the sluggishness to accumulated exposure to ozone has already been explained, and if necessary can easily be explained again without the use of equations

in the text. Mathematical expressions and integrals are not required at this point, but could rather be introduced earlier in the paper, for example in Chapter 2.

> We have moved the bulk of p6, lines 12-20 to instead be under section "Sluggishness parameter $\tau_{O_3}$ and modelled stomatal opening". This now means that in that section, not only is parameter $\tau_{O_3}$ defined, but also presented a discussion as to how it may change dependent on exposure time to elevated ozone levels.

P.7, lines 9-13: Repetition, and strictly not conclusions based on results of this article. Perhaps better placed under discussion.

> We have moved these sentences back to the Discussion part of the paper.

Technical corrections

P. 2, line 6: remove "is"

> Done

P.3, line 18: remove "and"?

> Done

P. 6, line 17: Remove parenthesis around the concentration based critical threshold.

> We have removed the square brackets around the concentration (and similarly elsewhere).

P. 6, line 21-22: please rephrase for clarity.

> We have re-worded the entire paragraph, as revisiting this text, it is clear there is also a lack of clarity between the points made in the three sentences. We now write: "If the ABA signalling process plays a key role in linking tropospheric ozone levels to stomata sluggish effects, then careful analysis is needed of data from experimental examples of well-watered vegetation at high ozone levels. This is because high ABA concentrations generally increase during periods of soil moisture stress, to which stomata response by lowering their opening. If therefore sluggishness is also observed during well-watered periods, and hence for low ABA concentrations, then this suggests that additional mechanisms operate beyond this hormone in linking ozone concentrations to inertial of stomata."

**Associate Editor Report**

Many thanks for your responses, which address the reviewer concerns sufficiently. I'm still worried that the evidence base for your proposed scheme is extremely thin. I would recommend to be more explicit about this in your conclusion section by revising the sentence "Targeted measurement campaigns may provide more detailed information on the appropriateness of our $\tau_{O3}$ formulation." to acknowledge that the application in an ESM context requires more data for parameterisation.

Thank you for this comment. We have re-written extensively the last paragraph, as repeated below. We make it clear that whilst the broad structure of our equation for sluggishness might still be valid, more measurement campaigns are needed to verify if it has exactly the correct format and dependencies. For instance, we note that there may be some nonlinearity in response to the gap between calculated stomatal conductance with/without $O_3$ damage. We also recognise that the strength of sluggishness – given by magnitude of $\tau_{O3}$ – may have a complex dependence on the history

of $O_3$ exposure levels. We hope the below provides the necessary level of caution, yet allows Eqn (1) to be a potential starting point for consideration in implementing "sluggishness" in to land surface models.

Targeted measurement campaigns may provide more detailed information on the appropriateness of our $\tau_{O3}$ formulation. This includes (a) whether this is a generic form for describing tropospheric ozone damage to vegetation (or alternatively, for instance, if the response may be nonlinear in $g_{l,slug} - g_l$), (b) how the $\tau_{O3}$ value depends on accumulated ozone exposure, or if there is a more complex dependence on $O_3$ exposure history, and (c) if there is potential to map on to broad PFTs. However if our formulation is broadly valid, then "sluggish" effects can be implemented within large-scale land surface models such as JULES (Clark et al., 2011) via our proposed Eq. (1). Furthermore, if valid, then eventual implementation in the largescale terrestrial models of ESMs offers hope that the implications of sluggish stomata can be understood in the context of simultaneous changing climatic conditions, the global carbon cycle and varying tropospheric ozone levels, along with any feedbacks.

---

## Author Response (AR2)

**CEH Wallingford**
Benson Lane, Wallingford, OXON, OX10 8BB,
U.K.
Email: chg@ceh.ac.uk
Tel: +44 (0)7884437138          August 28th 2018

The Editors
Biogeosciences Journal

Dear Editors,

Thank you for accepting our paper titled:

**Technical Note: A simple theoretical model framework to describe plant stomatal "sluggishness" in response to elevated ozone concentrations**

Uploaded is the "paper.tex", which is the LaTeX file, along with references in "paper.bib". The diagram is given within a .tar file.

We also supply a 500-character short summary of our analysis.

Thank you for all of your help so far with this paper. Please don't hesitate to contact me if you have any questions.

With kind regards,

Prof Chris Huntingford (and on behalf of co-authors)